# Patient Experience during Contrast-Enhanced Computed Tomography Examination: Anxiety, Feelings, and Safety

Sandra Lange [1], Wioletta Mędrzycka-Dąbrowska [2,*] and Anna Małecka-Dubiela [1]

1    Department of Internal and Pediatric Nursing, Medical University of Gdańsk, Dębinki 7,
     80-211 Gdańsk, Poland; langa94@gumed.edu.pl (S.L.); anna.malecka-dubiela@gumed.edu.pl (A.M.-D.)
2    Department of Anaesthesiology Nursing & Intensive Care, Faculty of Health Sciences,
     Medical University of Gdansk, Dębinki 7, 80-211 Gdańsk, Poland
*    Correspondence: wioletta.medrzycka@gumed.edu.pl

**Abstract:** Introductions: Computed tomography is one of the biggest breakthroughs in diagnostic imaging. In order to more accurately assess structures and pathological changes during the examination, it is necessary to administer a contrast agent. Patients presenting for the examination, very often only find out during the examination that a contrast agent is required. This increases patients' uncertainty when giving written consent for contrast administration, as well as anxiety about the examination. The aim of this study was to explore the experiences of patients who have contrast-enhanced CT scans, focusing primarily on anxiety, feelings, and safety. Methods: The cross-sectional study was conducted in diagnostic imaging offices in Pomeranian Voivodeship in 2019–2020. The survey was aimed at patients presenting for CT examinations with intravenous contrast injection. In total, 172 patients participated in the survey. A proprietary survey questionnaire was used to conduct the study. Results and Conclusions: During a CT scan, intravenous contrast agent administration is often necessary. Although there are few studies on patients' experiences with this examination, the authors observe that some patients experience anxiety. The results of our study showed the following: (1) 30.8% of patients experience anxiety before a CT scan with intravenous contrast injection; (2) variables such as gender, previous experience, and searching for information about the examination influence the occurrence of anxiety; (3) the most common feelings experienced by patients during intravenous contrast injection are a sensation of warmth spreading throughout the body; (4) the most common source of information about the study used among patients was the Internet; (5) most patients feel safe during a CT scan.

**Keywords:** contrast; CT examination; anxiety; feelings; safety

## 1. Introductions

Computed tomography (CT) is one of the biggest breakthroughs in diagnostic imaging [1]. CT scanning can be performed in the hospital setting, and in an emergency, as well as in an outpatient procedure, in order to make a diagnosis, narrowing down the differential diagnosis. In addition, CT can be used for screening, while performing a biopsy or during surgery [2]. In order to more precisely assess structures and pathological changes during the examination, a contrast agent is used [3,4]. Despite the unquestionable benefits that CT examination brings, it is associated with several direct risks that challenge radiology staff to ensure patient safety [5,6]. These include overexposure to radiation, contrast-induced allergic reactions, and post-contrast nephropathy. In addition, it is very important to ensure the mental and physical well-being of the patient who is under the care of the diagnostic team [6]. More than 70 million radiological examinations are performed worldwide using contrast agents [7]. The administration of a contrast agent, as in the case of drugs, is associated with the possibility of side effects [7,8]. Extrarenal acute adverse reactions associated with intravenous contrast administration are categorized as mild (mild

edema, urticaria, and nasal leakage), moderate (pruritus, bronchospasm, and facial edema) and severe (anaphylactic shock, facial and laryngeal edema, and bronchospasm with hypoxia) [5]. The administration of a contrast agent can also cause physiological reactions that do not require intervention and treatment. These include nausea, vomiting, anxiety, headache, cardiac arrhythmia, and chest pain, among others [5].

A basic legal and ethical requirement is to obtain informed consent from the patient before proceeding with an examination or treatment. The patient becomes part of the diagnostic and treatment process, which includes making decisions about their care [6]. The peri-radiological process begins as soon as the referring physician writes a referral for examination [9].

In the radiology department, a team including radiologists, nurses and electroradiology technicians is responsible for patient care before, during, and after the examination, and for ensuring safety [6,8]. In addition to cooperating with each other, each member of the team should be knowledgeable about their own roles and responsibilities with regard to ensuring patient safety [6].

Patient care begins the moment the patient arrives at the diagnostic office. In Poland, the nurse working in the CT scan unit is responsible for identifying the patient and checking their preparedness for the examination, reviewing their medical history with special attention to aspects that may constitute a contraindication to the examination and the administration of the contrast agent, such as, for example, abnormal renal indices, hyperthyroidism, previous contrast hypersensitivity reactions, or pregnancy. If there are any uncertainties, the nurse reports them to the radiologist, who decides on continuing the examination. Above that, the nurse is responsible for preparing the automatic contrast administration equipment. The next stage of preparation is the insertion of a peripheral cannula, in the appropriate location and of an appropriate size depending on the type of examination. Before administering a contrast agent, the cannula's patency must be checked by flushing it with saline. Although the patient remains alone in the examination room for the duration of the scan, the nurse constantly monitors their condition, ensures their safety and monitors the possibility of signaling the occurrence of worrying symptoms. After the examination, the inpatient patient returns to the ward with assistance from medical staff. Outpatients remain in the office area for observation. Another task of the radiology nurse is to coordinate and cooperate with the staff of other departments. They serve as a liaison in the flow of information and education to ensure optimal patient care and fluidity in the performance of examinations [10].

The authors' observations show that a large group of patients are not informed by referring physicians what the examination is about and what the contrast agent is. Patients presenting for the examination, very often learn only during the examination about the need to administer a contrast agent. This increases patients' uncertainty when giving written consent for contrast administration, as well as their fear of the examination.

A study among patients undergoing a magnetic resonance imaging (MRI) scan, found that examination-related anxiety occurs in 30%. Reasons for MRI examination anxiety include uncertainty and lack of knowledge about the situation (vigilance), fear of the MRI machine itself, anxiety about the test results, uncertainty due to lack of information, or agitation due to the circumstances of undergoing an MRI (e.g., the narrow space and lack of mobility) [11]. In contrast, there are few studies that focus on the anxiety and feelings of patients who undergo CT scans with intravenous contrast agent injection.

The aim of this study was to explore the experiences of patients who have contrast-enhanced CT scans, focusing primarily on anxiety, feelings, and safety. We answered the following questions: (1) Do patients experience anxiety before contrast-enhanced CT examination? (2) What factors influence the occurrence of anxiety before CT examination with contrast administration? (3) What feelings are most common in patients during intravenous contrast administration and what do they manifest as? (4) From what sources do patients most often learn about the examination? (5) Do patients feel safe during contrast CT examination?

## 2. Materials and Methods

### 2.1. Study Design

This study was a cross-sectional study. The survey was conducted in three diagnostic imaging offices in the Pomeranian region (Poland) in 2019–2020. In total, 172 patients participated in the study. The questionnaire was addressed to patients presenting to the institution for a CT scan with intravenous contrast injection.

### 2.2. Respondents and Procedure

Written permission was obtained from the management to conduct the study. Adult patients undergoing CT scans with contrast administration were included in the study, regardless of the area examined, and able to complete the questionnaire on their own. Participation in the study was voluntary, and the completion of the questionnaire was considered informed consent to participate in this study. Patients who were underage and inpatients or patients from emergency departments or with consciousness or vision disorders that made it impossible for them to read or complete the questionnaire were automatically excluded from the study.

### 2.3. Research Instruments

For the study, a 14-item questionnaire was created based on the researchers' clinical experience, as well as an analysis of the literature. The questionnaire consisted of two parts. The first part contained sociodemographic questions about gender, age, education, and place of residence. The second part consisted of closed-ended, single-choice questions to which patients answered yes or no (6 questions) and closed-ended questions with a definable number of answers (4 questions), which were about the source of knowledge about the examination, the medical staff, and the patient's feelings during contrast administration.

### 2.4. Data Analysis

Data analysis was conducted using Microsoft Excel and STATISTICA version 13 (Dell Statistica, Krakow, Poland). Data from the questionnaires are presented in tabular form. The numbers of respondents belonging to selected categories are given and presented in terms of the percentage of the total group. An analysis of the interdependence of qualitative characteristics was conducted using the $\chi^2$ (Chi-square) test. Results were considered statistically significant when $p \leq 0.05$.

To determine which variables are significantly related to the experience of anxiety before the CT scan, an effort was made to construct a logistic regression model.

### 2.5. Statement of Ethics

This study was approved by the Bioethics Committee of Medical University of Gdansk (Resolution no. NKBBN/31/2020). The researchers followed the appropriate legal rules and bioethical principles of the Declaration of Helsinki, giving due regard to the Strengthening the Reporting of Observational Studies in Epidemiology (STROBE) criteria.

## 3. Results

### 3.1. Characteristics of the Study Group

In total, 172 persons participated in the study, including 95 women (55.2%) and 77 men (44.8%). All respondents were over the age of 18. The largest group of respondents was of those aged 66 and over—32.5%. Table 1 presents the socio-demographic data of the respondents.

**Table 1.** Socio-demographic data.

| | Data | N | % |
|---|---|---|---|
| Gender | Female | 95 | 55.2 |
| | Male | 77 | 44.8 |
| Age | 18–35 | 23 | 13.4 |
| | 36–55 | 45 | 26.2 |
| | 56–65 | 48 | 27.9 |
| | 66 and over | 56 | 32.5 |
| Education | Primary education | 17 | 9.9 |
| | Secondary education | 122 | 70.9 |
| | Higher education | 33 | 19.2 |
| Residence | Village | 38 | 22.7 |
| | City with up to 60 thousand inhabitants | 56 | 32.6 |
| | City with more than 60 thousand inhabitants | 78 | 45.3 |

*3.2. Contrast-Enhanced CT Examination and Experience of Anxiety*

Of the 172 participants, 53 (30.8%) felt anxious before the examination, as shown in Table 2.

**Table 2.** Anxiety before CT examination with contrast administration.

| Anxiety Before CT Examination | N | % |
|---|---|---|
| I have not felt anxiety | 119 | 69.2 |
| I have felt anxiety | 53 | 30.8 |
| Total | 172 | 100 |

Based on the responses, feelings of anxiety before the study were analyzed in relation to other variables. The results of the analyses are presented below.

3.2.1. Variable: Gender

The analysis showed that there was a significant correlation between feeling anxiety before the examination and the gender of the examined person ($\chi^2 = 10.4$, $p = 0.001$). Among women, anxiety before the examination was felt by more than 4 in 10 subjects. In men, the percentage was significantly lower, affecting less than 2 in 10 subjects. Table 3 presents a summary of the collected results.

**Table 3.** Anxiety before examination in terms of gender of respondents.

| Gender | Anxiety Before CT Examination | | Total |
|---|---|---|---|
| | No | Yes | |
| Female | 56 | 39 | 95 |
| % | 59.0% | 41.0% | |
| Male | 63 | 14 | 77 |
| % | 81.8% | 18.2% | |
| Total | 119 | 53 | 172 |

3.2.2. Variable: Age

Each person was assigned to one of four age categories: 18–35 years, 36–55 years, 56–65 years, and 66 years and older. The analysis showed no significant correlation between

experiencing anxiety before the survey and the respondent's membership in one of the four age categories.

### 3.2.3. Variable: Previous Experience

Respondents were asked if this was their first CT scan. An analysis of the correlation between feeling anxious before the examination and having experience of previous CT examinations showed a significant relationship ($\chi^2 = 6.2$, $p = 0.013$), which is presented in Table 4.

**Table 4.** Anxiety before the examination in terms of having previous experience with CT scanning.

| Was This Your First CT Scan? | Anxiety Before CT Examination | | Total |
| --- | --- | --- | --- |
| | No | Yes | |
| No | 80 | 25 | 105 |
| % | 76.2% | 23.8% | |
| Yes | 39 | 28 | 67 |
| % | 58.2% | 41.8% | |
| Total | 119 | 53 | 172 |

In the group of people for whom the CT examination was the first, more than 4 in 10 felt anxious before the examination. In the group of people for whom it was not the first CT scan, this percentage was significantly lower, affecting just over 2 in 10 patients.

### 3.2.4. Variable: Searching for Information about the Examination

Respondents were asked whether or not they searched for information about CT before the examination. An analysis of the correlation between feeling anxious before the examination and searching for information about it showed a significant correlation ($\chi^2 = 11.6$, $p < 0.001$), which is shown in Table 5.

**Table 5.** Anxiety before the examination in terms of the respondent's previous search for information about it.

| Did You Search for Information About CT Scans Before the Examination? | Anxiety Before CT Examination | | Total |
| --- | --- | --- | --- |
| | No | Yes | |
| No | 76 | 19 | 95 |
| % | 80.0% | 20.0% | |
| Yes | 43 | 34 | 77 |
| % | 55.8% | 44.2% | |
| Total | 119 | 53 | 172 |

In the group of people who searched for information about it before the examination, anxiety was felt by more than 4 out of 10 respondents, while in the group of people who did not search for such information the percentage was significantly lower—anxiety was felt by 2 out of 10 respondents.

### 3.3. Contrast-Enhanced CT Examination and the Sensations Accompanying Its Administration

The most common sensation accompanying the administration of contrast, both among women and men, was a feeling of warmth spreading throughout the body. Briefly, 80 female respondents (84.2%) and 68 male respondents (88.3%) indicated this sensation. The next most frequently indicated sensation among women was the feeling of pressure on the bladder, with 61 persons (64.2%), while in men this was the sensation of a metallic taste in

the mouth, with 22 persons (28.6%). Briefly, 39 female respondents (41.1%) indicated that they experienced a sensation of a metallic taste in the mouth during contrast administration. The sensation of pressure on the urinary bladder was experienced by 17 male respondents (22.1%). Other sensations were rarely indicated by the subjects, accounting for less than 10% of the responses, and were more often experienced among women. A summary of all the sensations during the administration of a contrast agent is shown in Table 6.

**Table 6.** Feeling during administration of contrast injection.

| Feelings | Gender | | | |
|---|---|---|---|---|
| | Female | | Male | |
| | *n* | % | *n* | % |
| Warmth throughout the body | 80 | 84.2 | 68 | 88.3 |
| Pain | 2 | 2.1 | 0 | 0 |
| A metallic taste in the mouth | 39 | 41.1 | 22 | 28.6 |
| Dizziness | 8 | 8.4 | 2 | 2.6 |
| Pressure on the bladder | 61 | 64.2 | 17 | 22.1 |
| Nausea | 3 | 3.2 | 0 | 0 |
| Heart palpitations | 4 | 4.2 | 0 | 0 |
| Others | 4 | 4.2 | 7 | 9.1 |

Above that, the respondents were asked in the questionnaire to identify a member of medical staff from whom they received information about the possibility of the above feelings and how they should behave after a contrast-enhanced CT scan. Most respondents indicated a nurse as a source of information and this amounted to 85.9% of respondents. An electroradiology technician was indicated for 6.5% of responses, and a radiologist for 5.3%. The doctor who referred the patient for the examination was the least likely to provide information (1.8%).

### 3.4. Sources of Knowledge about the Examination among Patients

Of all respondents, 82 patients (47.7%) had searched for information about it before the examination. As the most common source of knowledge, patients pointed to the Internet. They accounted for 60 of the respondents (73.2%). Another source of knowledge was the patients' friends—15 respondents (18.3%). Only four respondents (4.9%) gained knowledge from information brochures. Three respondents (3.6%) used other sources.

### 3.5. Patients' Feeling of Safety during CT Examination

The analysis showed that 96.8% of women and 100% of men felt safe during the examination. Briefly, 3.2% of female respondents indicated a lack of feeling of safety. Most respondents confirmed that the presence of a nurse in the cabinet increased their feeling of safety. Among women, this amounted to 91.6% of respondents, and among men, this percentage was 83.1%.

## 4. Logistic Regression Model

Analyses showed that the variables age of the respondents and education were not found to be significant. The final logistic regression model consists of three variables that remain significantly related to feelings of anxiety before the study. Details of the logistic regression are shown in Table 7 below.

**Table 7.** OR and test probability values for variables in a logistic regression model modeling the experience of anxiety before a CT scan.

| Result | | Perception the Fear Before Examination—Odds Ratios (OR) Modeled Probability: Feeling Anxious Before Examination = Yes | | | |
|---|---|---|---|---|---|
| | | **OR** | **95% CI for OR** | **95% CI for OR** | ***p*-Value** |
| | | | **Upper** | **Lower** | |
| Gender | Female | 2.59 | 1.24 | 5.44 | 0.011 |
| First CT scan | Yes | 2.07 | 1.03 | 4.17 | 0.040 |
| Pre-CT scan information search | Yes | 2.54 | 1.25 | 5.13 | 0.010 |

Due to previous analyses for the variables gender, first CT scan and pre-examination information searching, those trait values for which pre-examination anxiety was significantly lower, i.e., male (for the gender variable), No (first CT scan) and No (pre-examination information searching), respectively, were established as reference levels. Each of the variables is significantly related to feeling anxious before a CT scan, and the determined adjusted odds quotients mean that women are almost 2.6 more likely than men to feel anxious before a scan. In those who search for information before the examination, the chance of feeling anxiety is more than 2.5 times higher than that in those who do not search for such information. In patients who undergo a CT examination for the first time, the chance of feeling anxiety is more than twice as high as in those who have undergone such an examination before.

## 5. Discussion

The last decade has seen a significant increase in the availability of diagnostic examinations and an expansion in the use of CT imaging [12]. At the same time, concern about patients' exposure to ionizing radiation and related complications has increased [13]. This has led to a number of innovations in the diagnostic sector aimed at reducing patient exposure to radiation doses while maintaining the high quality and efficiency of these examinations [6]. In contrast, few studies have been focused on assessing patients' awareness and knowledge of the examination and their feelings of safety [14]. The analysis showed that 30.8% of patients presenting for a CT scan with intravenous contrast injection felt anxious about it. Among them, 41% were women and 18.2% were men. As in a study conducted at Charles University and General University Hospital in Prague by Lambertova et al., the female gender was significantly more likely to report anxiety before the examination. The study also found that anxiety was more common in younger patients [14]. The self-analysis conducted did not show that the age of the patients significantly affects the feeling of anxiety before the examination.

The patients' previous experience of CT examination significantly influences the feeling of anxiety before it. The analysis showed that first-time patients attending the examination were significantly more likely to experience anxiety. They accounted for 41.8% of respondents. Among respondents for whom this was a subsequent examination, the percentage was significantly lower at 23.8%. Also in the study by Lambertova et al., the analysis showed that young women in particular, with no previous experience of CT scans, were more likely to feel anxious [14]. The study showed a significant correlation between searching for information about the examination and experiencing anxiety. Among those who searched for information about the examination before it, 34% felt anxious. The percentage of those who did not search for such information was lower, at 19%. However, these results should be interpreted with caution, as the study did not take into account the possibility that patients who did a search for information about the study were already anxious. In a study conducted by Lambertova et al., prior to the contrast-enhanced CT

procedure, patients completed a knowledge test about the procedure, and then received brochures containing basic information about the examination. The analysis of the results also showed that, in general, patients felt more fear after reading this information [14].

During intravenous contrast injection, the most common sensation reported by both men and women was a feeling of spreading warmth throughout the body. A study by Wyszomirska E. et al. that analyzed mild adverse events after the administration of iodine as a contrast agent confirms the result obtained in our study [4]. Another sensation that occurred in 41.1% of women was a metallic taste in the mouth. Among male respondents, this response was indicated by 28.6% of participants. The third most frequently indicated symptom was a feeling of pressure on the urinary bladder, which occurred in 64.2% of men and 22.1% of women. In contrast to their own study, in the analysis by Wyszomierska et al. women indicated a feeling of urination more often than men did [4].

The data collected show that 73.2% of patients use the Internet as a source of knowledge about the examination. In contrast, 18.3% of respondents rely on the opinions of friends. In the Scholz et al. study, while 76.9% of patients had already experienced the CT examination and related instructions, the procedure was new to 23.1% of patients. In total, 72.5% (n = 116) of all patients were not familiar with other information materials (patient brochures, websites, and television programs) prior to the examination [15]. In the study by Lambertova et al., only 8% of respondents used the Internet [14]. Above that, our study showed that in the diagnostic department, the radiology nurse is most often the person who educates patients. This was the response indicated by 85.9% of respondents. This is in contrast to data from a study by Caoili et al. which showed that the referring doctor was the main source of information about CT scans (47%) [16]. The literature points out that brochures, information sheets, and reliable online sources can be used by medical personnel to improve patients' knowledge of the examination, but stresses that this is no substitute for direct contact with a professional to ensure a correct understanding of the information [14,17,18]. Scholz et al., in their study, identified determinants of satisfaction related to communication with patients during check-in prior to performing contrast-enhanced CT scans. The study found that overall satisfaction with the briefing by the radiologist prior to the contrast-enhanced CT scan was high. The strongest predictor of patient satisfaction was the clarity of the briefing [15]. Our own survey shows that most patients surveyed feel safe during a CT scan with the intravenous injection of a contrast. They make up 98.2% of respondents.

Although the organization of work in diagnostic imaging departments may vary in different countries, the results of this study can be used. Regardless of which medical staff members (radiologists, electroradiology technicians, or radiology nurses) are responsible for direct contact with the patient, it is important to pay attention to the attitude of the medical staff, as it can affect the patient's attitude toward future examinations. Knowledge of the most common feelings reported by patients during contrast injection allows the patient to be warned in advance of their occurrence. It is also important to verify information about the examination contained on websites and patient information leaflets.

## 6. Conclusions

Contrast-enhanced CT is a commonly performed examination for the diagnosis and differentiation of many diseases. Often, the intravenous administration of a contrast agent is required during it. Although there are few studies on patients' experiences with this examination, the authors observe that some patients experience anxiety. The results of our study showed that (1) 30.8% of patients experience anxiety before a CT scan with intravenous contrast injection; (2) variables such as gender, previous experience, and searching for information about the examination influence the occurrence of anxiety; (3) the most common feelings experienced by patients during intravenous contrast injection are a sensation of warmth spreading throughout the body; (4) the most common source of information about the examination among patients is the Internet; (5) most patients feel safe during a contrast-enhanced CT scan.

The results provide insight into the experience of patients undergoing contrast-enhanced CT, but the limitations of the study must be taken into account in their interpretation.

## 7. Implications for Practice

Previous experience has proven to be a factor in experiencing anxiety before the examination. The attitude of the medical staff working in the diagnostic office is significant. Therefore, efforts should be made to maintain a high-quality relationship between the patient and the medical staff, for example, through regular communication training. Another aspect that affects the fear of the examination is the information obtained about it. Given that the Internet is a major source of knowledge, medical institutions should pay special attention to the information posted on their websites, ensure patient education about contrast-enhanced CT, and verify their knowledge. Knowledge of the most common sensations that intravenous contrast administration causes enables the patient to prepare for them in advance. In our opinion, the nurse is the right person to function as an educator in the radiology office. According to our study, the nurse is most likely to have direct contact with the patient in the CT office and to enhance the patient's feeling of safety. In order to prepare the patient and provide information related to the examination, leaflets or brochures that the patient could become familiar with at home may be helpful.

## 8. Limitations

Our study has some limitations. First, the study was conducted in three diagnostic imaging offices, but from the same provider, with the same procedures. Depending on the medical center, these may differ and thus affect the results. Therefore, the survey results cannot be generalized. Another limitation is the relatively small number of respondents. Multicenter studies with a larger number of participants are needed to confirm the results. The fact that the survey did not include which patients searched for information about the study also remains a limitation. The study did not determine whether respondents were anxious or whether searching for information made them anxious.

**Author Contributions:** Conceptualization, S.L., methodology, S.L. and A.M.-D.; formal analysis, S.L.; writing—original draft preparation, S.L., writing—review and editing, W.M.-D. and A.M.-D.; visualization, W.M.-D.; supervision, A.M.-D.; All authors have read and agreed to the published version of the manuscript.

**Funding:** This research received no external funding.

**Institutional Review Board Statement:** The study was conducted in accordance with the guidelines of the Declaration of Helsinki, and approved by the Bioethics Committee of the Medical University of Gdansk (resolution no. NKBBN/31/2020).

**Informed Consent Statement:** Not applicable.

**Data Availability Statement:** The authors declare that the data of this research are available from the correspondence author on request.

**Conflicts of Interest:** The authors declare no conflict of interest.

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
