# Peer review of "Patient Experience during Contrast-Enhanced Computed Tomography Examination: Anxiety, Feelings, and Safety"

_safety, 2023_

Round 1

Reviewer 1 Report

The study reports anxiety related to the use of contrast medium during the CT examination.

The paper is interesting, the data collected is accurate and suitable for the study.

I suggest that in the implications for practice, information leaflets can be used in addition to the nurse to inform patients.

Detect and correct a some minor grammar errors

Author Response

Reviewer 1

The study reports anxiety related to the use of contrast medium during the CT examination.

The paper is interesting, the data collected is accurate and suitable for the study.

I suggest that in the implications for practice, information leaflets can be used in addition to the nurse to inform patients.

Detect and correct a some minor grammar errors

Dear Reviewer 1,

thank you for the suggestion, which has been added to the implications for practice.

The changes have been highlighted in yellow.

Reviewer 2 Report

Dear authors,

Your manuscript is interesting but I need you to answer some questions:

INTRODUCTION

-          The introduction should not be to have subsections.

-          The authors have not stated the objective of the investigation. An objective is not about answering questions.

-           

MATERIALS AND METHODS

Study Design:

-          What was the target population? How was the sample chosen? The authors must specify it.

Research Instruments:

-          Has the self-developed scale been validated? The authors should provide evidence of the validity of the scale.

-          In addition, the authors should provide a copy of the scale and explain how each item is rated.

RESULTS

-          The authors must include the response rate of the participants in the study.

-          The results are very extensive. You must eliminate redundant or uninteresting information.

DISCUSSION

-          The authors have not explained how to apply the research to clinical practice.

-          The authors have not included study limitations.

CONCLUSIONS

-          In a cross-sectional study, cause-effect relationships cannot be established. Conclusions should be more cautious.

REFERENCES

-          Many bibliographies are obsolete. The bibliographic citations used are more than 5 years old (57.7 %). The authors must update and arrange the bibliography.

Author Response

Reviewer 2
Dear authors,
Your manuscript is interesting but I need you to answer some questions:

INTRODUCTION

-          The introduction should not be to have subsections.
-          The authors have not stated the objective of the investigation. An objective is not about answering questions.

MATERIALS AND METHODS

Study Design:
-          What was the target population? How was the sample chosen? The authors must specify it.

Research Instruments:
-          Has the self-developed scale been validated? The authors should provide evidence of the validity of the scale.
-          In addition, the authors should provide a copy of the scale and explain how each item is rated.
RESULTS
-          The authors must include the response rate of the participants in the study.
-          The results are very extensive. You must eliminate redundant or uninteresting information.

DISCUSSION
-          The authors have not explained how to apply the research to clinical practice.
-          The authors have not included study limitations

CONCLUSIONS
- In a cross-sectional study, cause-effect relationships cannot be established. Conclusions should be more cautious.

REFERENCES
- Many bibliographies are obsolete. The bibliographic citations used are more than 5 years old (57.7 %). The authors must update and arrange the bibliography.

Dear Reviewer 2,

Thank you for your insightful review and all your suggestions. Changes and clarifications have been made to improve the quality of the manuscript and are indicated in blue.

INTRODUCTION:
- The introduction has been rewritten.
-
The aim of the study was formulated.

MATERIALS AND METHODS
Study Design:
- Information was added to Study desing section.
Research Instruments:
- Section reformulated. The English translation of the questionnaire is in Appendix.

RESULTS
-The response rate is impossible to determine.
- Revisions were made, the number of tables was reduced.

DISCUSSION
- Implications for clinical practice and limitations of the study are included in separate sections (7. Implications for practice and 8. Limitations)

CONCLUSION
- This was emphasized in the conclusion, in the discussion, and noted in the limitations.

REFERENCES
-The bibliography has been updated.
Bibliographic citations up to 5 years = 72.2%; up to 10 years = 22.2% and one older than 10 years.

Reviewer 3 Report

Congratulations to the researchers, but I need a series of improvements to better understand the article, the use of acronyms in the title should be omitted. I cannot evaluate an article whose data was obtained more than 3 years ago, it generates a significant bias, they should consider reviewing and updating the data, as happens with bibliographic sources that need a thorough review.

Author Response

Reviewer 3

Congratulations to the researchers, but I need a series of improvements to better understand the article, the use of acronyms in the title should be omitted. I cannot evaluate an article whose data was obtained more than 3 years ago, it generates a significant bias, they should consider reviewing and updating the data, as happens with bibliographic sources that need a thorough review.

Reviewer 3

Thank you for your review. The acronym in the title has been replaced. A series of corrections have been made to the paper and clarifications have been added. The bibliography has also been updated.

Reviewer 4 Report

Interesting research

Author Response

Reviewer 4

Interesting research

Dear Reviewer 4,

Thank you for your positive review.

Reviewer 5 Report

-        Introduction: The whole section needs to be more specific for the topic. General aspects of radiology do not need to be described in that detail. Furthermore, general aspects on patient safety are not needed here.

-        Methods: Chapter 2.1 is much too short and needs to be combined with another chapter.

-        Methods, Chapter 2.4: You should write “Microsoft Excel”.

-        Methods, Chapter 2.4: What is meant by “verified” here?

-        Results: It is a purely descriptive study, although one regression model is presented. It includes too many tables and offers only very limited insights.

-        Table 10: The description of the CI is wrong.

-        Abstract: The last paragraph is only on results and not conclusions as outlined.

-        Lines 46-47: This is a sentence written in Polish.

Author Response

Reviewer 5

-        Introduction: The whole section needs to be more specific for the topic. General aspects of radiology do not need to be described in that detail. Furthermore, general aspects on patient safety are not needed here.

-        Methods: Chapter 2.1 is much too short and needs to be combined with another chapter.

-        Methods, Chapter 2.4: You should write “Microsoft Excel”.

-        Methods, Chapter 2.4: What is meant by “verified” here?

-        Results: It is a purely descriptive study, although one regression model is presented. It includes too many tables and offers only very limited insights.

-        Table 10: The description of the CI is wrong.

-        Abstract: The last paragraph is only on results and not conclusions as outlined.

-        Lines 46-47: This is a sentence written in Polish.

Dear Reviewer 5,

Thank you for your suggestions, which have been taken into account to improve the quality of the manuscript. Changes are marked in orange.

- Introduction: has been rewritten. Unnecessary aspects have been removed, and subsections have been combined.

- Methods, Chapter 2.4: corrected.

-Methods, Chapter 2.4: sentence reformulated.

-Results: was revised and the number of tables reduced.

-Table 10: corrected.

- Abstract: corrected, both in the abstract and in the conclusions section.

-Lines 46-47: sentence translated.

Reviewer 6 Report

This manuscript describes patients’ anxiety and feelings during contrast-enhanced CT using questionnaire.

 The strength of the paper is clear presentation of the outcomes and adequate sample size for chi-square statistical test.

 The weakness of the study is limited support for causal relationship. Discussion on the effect of pre-CT information search is difficult. It is not clear whether worried subjects performed the search or search added worry to the subjects. This came from the study design and should be regarded as limitation of the study rather than flaw.

Points to be considered:

1)     As this is a local study, adding concise consideration on the potential extrapolation to another country would add value to the study. The support for extrapolation would be similarity of medical standard among the country.

2)     Line 131: In Study Design section, comprehension of study design is warranted. Sample size and number of medical centers should be addressed in addition to cross sectional nature.

3)     Table 7, Line 297, 330: Warmth and heal should be distinguished.

Author Response

Reviewer 6

This manuscript describes patients’ anxiety and feelings during contrast-enhanced CT using questionnaire.

 The strength of the paper is clear presentation of the outcomes and adequate sample size for chi-square statistical test.

 The weakness of the study is limited support for causal relationship. Discussion on the effect of pre-CT information search is difficult. It is not clear whether worried subjects performed the search or search added worry to the subjects. This came from the study design and should be regarded as limitation of the study rather than flaw.

Points to be considered:

1)     As this is a local study, adding concise consideration on the potential extrapolation to another country would add value to the study. The support for extrapolation would be similarity of medical standard among the country.

2)     Line 131: In Study Design section, comprehension of study design is warranted. Sample size and number of medical centers should be addressed in addition to cross sectional nature.

3)     Table 7, Line 297, 330: Warmth and heal should be distinguished.

Dear Reviewer 6,

thank you for your suggestions. To improve the quality of the study, we have taken them into account in our manuscript.
The changes have been highlighted in green.

Points to be considered:

1) It was added at the end of the discussion.

2) Information has been added to Study Design section.

3) Has been distinguished. (Warmth throughout the body)

Round 2

Reviewer 2 Report

Dear authors,

Thanks for your reply. The explanations that you provide are satisfactory. The paper has greatly improved its quality.

Congratulations on your work.

Best regards

Reviewer 3 Report

Dear authors, the requested changes have been made.

Reviewer 5 Report

My specific comments have been addressed.

Reviewer 6 Report

The manuscript was adequately revised.